A novel perceptual two layer image fusion using deep learning for imbalanced COVID-19 dataset

http://orcid.org/0000-0001-5409-1305 Elzeki Omar M. 1 omar_m_elzeki@mans.edu.eg
http://orcid.org/0000-0003-2390-5665 Abd Elfattah Mohamed 2
http://orcid.org/0000-0002-8714-567X Salem Hanaa 3 hana.salem@deltauniv.edu.eg
http://orcid.org/0000-0002-9989-6681 Hassanien Aboul Ella 4 5
http://orcid.org/0000-0003-3021-5902 Shams Mahmoud 6
1 Faculty of Computers and Information Sciences, Mansoura University , Mansoura , Egypt
2 Misr Higher Institute for Commerce and Computers , Mansoura , Egypt
3 Communications and Computers Engineering Department, Faculty of Engineering, Delta University for Science and Technology , Gamasa , Egypt
4 Faculty of Computers and Artificial Intelligence, Cairo University , Cairo , Egypt
5 Scientific Research Group in Egypt (SRGE) , Cairo , Egypt
6 Faculty of Artificial Intelligence, Kafrelsheikh University , Kafrelsheikh , Egypt
Damaševičius Robertas
Electronic publication date: 2021 Feb 10
Publication date: 2021
Volume: 7
Electronic Location ID: e364
Received 2020 Oct 22; Accepted 2020 Dec 30
Copyright: © 2021 Elzeki et al.
Copyright year: 2021
Copyright holder: Elzeki et al.
License: This is an open access article distributed under the terms of the Creative Commons Attribution License, which permits unrestricted use, distribution, reproduction and adaptation in any medium and for any purpose provided that it is properly attributed. For attribution, the original author(s), title, publication source (PeerJ Computer Science) and either DOI or URL of the article must be cited.
License URL: https://creativecommons.org/licenses/by/4.0/

Keywords: Coronavirus, COVID19, NSCT, CNN, Feature extraction, Feature analysis, Image fusion, Deep learning, Machine learning, VGG19

Funding: The authors received no funding for this work.

==============================
Background and Purpose

COVID-19 is a new strain of viruses that causes life stoppage worldwide. At this time, the new coronavirus COVID-19 is spreading rapidly across the world and poses a threat to people’s health. Experimental medical tests and analysis have shown that the infection of lungs occurs in almost all COVID-19 patients. Although Computed Tomography of the chest is a useful imaging method for diagnosing diseases related to the lung, chest X-ray (CXR) is more widely available, mainly due to its lower price and results. Deep learning (DL), one of the significant popular artificial intelligence techniques, is an effective way to help doctors analyze how a large number of CXR images is crucial to performance.

Materials and Methods

In this article, we propose a novel perceptual two-layer image fusion using DL to obtain more informative CXR images for a COVID-19 dataset. To assess the proposed algorithm performance, the dataset used for this work includes 87 CXR images acquired from 25 cases, all of which were confirmed with COVID-19. The dataset preprocessing is needed to facilitate the role of convolutional neural networks (CNN). Thus, hybrid decomposition and fusion of Nonsubsampled Contourlet Transform (NSCT) and CNN_VGG19 as feature extractor was used.

Results

Our experimental results show that imbalanced COVID-19 datasets can be reliably generated by the algorithm established here. Compared to the COVID-19 dataset used, the fuzed images have more features and characteristics. In evaluation performance measures, six metrics are applied, such as QAB/F, QMI, PSNR, SSIM, SF, and STD, to determine the evaluation of various medical image fusion (MIF). In the QMI, PSNR, SSIM, the proposed algorithm NSCT + CNN_VGG19 achieves the greatest and the features characteristics found in the fuzed image is the largest. We can deduce that the proposed fusion algorithm is efficient enough to generate CXR COVID-19 images that are more useful for the examiner to explore patient status.

Conclusions

A novel image fusion algorithm using DL for an imbalanced COVID-19 dataset is the crucial contribution of this work. Extensive results of the experiment display that the proposed algorithm NSCT + CNN_VGG19 outperforms competitive image fusion algorithms.

Introduction

MIF provides crucial information representing the source images helpful in diagnosis, prognosis, treatment, and classification (Ganasala & Kumar, 2016). For a quick and accurate diagnosis, supplementary information must be extracted from the various source images in one image. It is well known that medical images have different and variable modalities that carry information with complementary properties (Srivastava, Prakash & Khare, 2016). A selection of image pixels or patches is performed to construct a fuzed image in the spatial domain to preserve each source image’s information. The major limitation of spatial information fusion is the non-integrality of the fuzed information producing the contrast and sharpness, which in turn leads to the decrease of detailed information in the fuzed image (Liu et al., 2017a; Zhu et al., 2018, 2019).

Chest X-Ray imaging, also called CXR, is generally a common method and non-invasive radiology examination. In the recent pandemic of COVID-19, the use of CXR is desired because radiologists easily interpret it, and the time-consumption is decreased with minimum assessment errors (Liu et al., 2019). The fusion of CXR images can be performed in feature extraction and classification stages, as demonstrated by Liu et al. (2019) and Huang et al. (2020a). Furthermore, CXR can be utilized as large-scale input images that can be combined with deep convolution neural networks (DCNN) to boost the performance of the variable sizes of thoracic diseases (Hu et al., 2020)

One of the recent fusion techniques is fusion based on Nonsubsampled Contourlet Transform (NSCT). The fusion technique can avoid spectral aliasing and provide more characteristics of the invariance translation (Huang et al., 2020b). Moreover, NSCT is proposed in the input image decomposition level to transform the source image to both low and high-pass subbands, providing more details and reservation of input images (Liu et al., 2019). The limitation of using NSCT is the resulting fusion performance in humanoid visualization based objective metrics that need more enhancement like adding an optimizer or classifier (Bhatnagar, Wu & Liu, 2013; Tian, Yibing & Fang, 2016; Gomathi & Kalaavathi, 2016). Another fusion technique is the utilization of deep learning in fusion strategies. Liu et al. (2018) present convolutional neural networks (CNN) for image fusion by which a weighted map of the source images are generated with promising results.

The correlation between NSCT and deep learning is demonstrated in Liu et al. (2017b) by which a fusion based on CNN and NSCT of multi-focus images is performed. Moreover, two-scale decomposition transforms are presented in Lahoud & Süsstrunk (2019) such that the image layers are fuzed based on CNN intermediate feature maps. They used a guided filter to smooth the weight maps and enforce consistency with the source images. Therefore, the merge between deep learning and NSCT is very attractive and helpful in discovering more details and consuming a minimum of less time (Hermessi, Mourali & Zagrouba, 2018; Tang et al., 2018; Amin-Naji, Aghagolzadeh & Ezoji, 2019).

The development and design of an effective MIF algorithm based on DL is still an open area. The key contributions of this research may be summarized as the following:Initially, the proposed approach decomposes the image into subbands using NSCT.

For feature extraction of the output CXR COVID-19 images from NSCT, CNN-VGG19 is then utilized.

Euclidean distance and weights subband calculations are applied to obtain the fuzed rules, a temporal consistency of extracted features, Euclidean distance, and weights subband calculations.

A fuzed image is computed using the inverse NSCT.

Finally, the comparative evaluation was performed using two methods; the first method is to determine the pre-trained framework efficiency using evaluation metrics. While the second method is based on classifying the fuzed CXR COVID-19 images using the deep learning approach CNN-VGG19 compared with the state-of-the-art.

The paper is organized as follows: in “Related Work”, the authors review the research field's literature. “Proposed Algorithm Framework” displays the suggested algorithm design, evaluation measures, and implementation techniques. The discussion and results then follow in “Evaluation Matrices” and “Experimental Results and Discussion”, and the conclusion of the research is then stated in “Conclusion”.

Related Work

Before explaining our architecture in more depth, this article's following section presents a brief introduction to the NSCT fusion strategy and deep neural network.

COVID-19 or coronavirus is an updated version of pneumonia of unknown cause found in Wuhan, China, and was first confirmed in China’s WHO Country Office; the disease was named COVID-19 by WHO (Cascella et al., 2020). To fight this virus’s spread, the cooperation between specialists in medical and artificial intelligence is required. From this pandemic, the attempts to diagnosis, classify, detect, and specify the suitable recovery method is performed and widely spread all over the countries (Bullock et al., 2020; Shi et al., 2020; Pham et al., 2020; Elavarasan & Pugazhendhi, 2020; Vafea et al., 2020; Raoofi et al., 2020).

Certainly, data fusion is essential to discover more details and improve observed data's extracted features (Meng et al., 2020; Attallah, Sharkas & Gadelkarim, 2020; Thabtah & Peebles, 2020). Typically, the Generative Adversarial Network (GAN) is widely used for data augmentation, especially for small data presented by Shams et al. (2020) for CXR images. In medical applications, the fusion of images is performed to discover essential parts (Tian, Yibing & Fang, 2016). It is well known that images are in three levels: binary level 0 and one, grayscale level 0 to 255, and RGB level. Most medical images, especially CT and CXR images, are grayscale (Ran et al., 2020).

Image fusion (IF) is an essential branch of information science. IF was widely used in different fields, including medical imaging, bioinformatics, simulation of military targets, etc. (Raol, 2009). This paper will study the fusion strategies based NSCT, deep learning CNN-VGG19, and the hybrid of NSCT and deep learning CNN-VGG19.

NSCT fusion strategy

The combination of low and high-frequency coefficient of source images is called NSCT (Bhatnagar, Wu & Liu, 2013) presented an architecture applied to CT and MRI images to be fuzed using NSCT to extract edge information and prominent texture based on directive contrast of the frequency coefficients. The main limitation is the shift variance problem that may occur in the fuzed images. To overcome this problem, a cascaded combination of NSCT and stationary wavelet transform (SWT) is presented by Bhateja et al. (2015) to enhance the phase's shift variance problem information of the fuzed images.

Precisely, pixel-level image fusion is promising in many image fusion strategies. Li et al. (2017) present multi-scale transformation coefficients to produce fuzed images with inter-scale correlation. They apply MRI and PET images to observe fusion performance objectives and determine the source images’ miss-registration.

Hybrid decomposition of NSCT and morphological sequential toggle operator (MSTO) is presented by Wang et al. (2020). Their methodology can extract significant feature information of the source images while preserving the unambiguous edges with a little produced noise in both visible and infrared image fusion.

Bashir et al. (2019) present an algorithm for multi-modal imagery based on SWT, and principal component analysis (PCA) applied to CXR, CT, and MRI images.

In addition to using NSCT in fusion strategy, the noise distribution produced by CXR images can be handled using Poisson-Gaussian noise analysis, as presented by Lee, Lee & Kang (2018). The authors apply their algorithm on CXR images to reduce the resulting noise images. Moreover, Chandra et al. (2020) presented an algorithm to extract shape features from CXR images based on a gray level co-occurrence matrix (GLCM) with an improved abnormality detection.

Deep learning in image fusion strategies

Deep learning (DL) approaches can be used as a late step in most fusion strategies (Lee, Mohammad & Henning, 2018). Most of CT and CXR images in medical applications can be handcrafted and fuzed in score level fusion strategy (Baumgartl et al., 2020).

Moreover, DL can be used as a feature extractor by which the fusion process is carried out in the feature extraction step. Next, the choice of features is determined using both CNN and PCA presented by Bhandary et al. (2020). The combination of one-dimensional feature vectors and the dimensionality reduction is performed using PCA are then applied to the source CXR images and tested the normal bacterial pneumonia.

A proposed method based on pre-trained CNN to fuze different subsets and transfer learning classifiers is presented by Ozkaya, Ozturk & Barstugan (2020). It is applied to CT images to classify COVID-19 cases. To visualize different registration of essential data of the source images using fixed and moving data labels as well as fixed and moving images, Haskins, Kruger & Yan (2020) presented an algorithm based on DCNN applied in MRI, CXR, and CT modality.

Hybrid NSCT and CNN fusion strategies

Hybrid techniques are generally helpful in many medical applications as it supports the strength points of approach while avoiding issues of shortage (Jaradat & Langari, 2009). In this study, we implement the use of a hybrid fusion technique using NSCT and CNN. The fusion of infrared and visible medical images can be performed using NSCT and dual-channel of pulse coupled neural network (PCNN) as presented by Xiang, Yan & Gao (2015).

A deep-stacked convolution neural network (DSCNN) for multiband images represent CT, MRI, and PET scan are presented by Lin et al. (2020). They used DSCNN and NSCT to fuze multiband images reconstructed by long short-term memory (LSTM) and DSCNN to overcome the data-driven approach's controllability problem.

Hybrid multimodality medical image fusion applied in both CXR and CT images were presented by Rajalingam, Priya & Bhavani (2018) based on convolutional and hybrid algorithms for disease analysis. A fusion architecture based on CNN of two source images and decomposition level based on NSCT is reconstructed using Gaussian pyramid reconstruction of the fuzed images (Huang et al., 2020b).

The maximum selection fusion rule of two source CT images based on NSCT and spatial frequency analysis of the source images are applied to the pulse coupled neural network presented as hybrid fusion architecture by Das & Kundu (2012). Moreover, for MRI modality (Maharjan et al., 2020) proposed a hybrid model to detect brain tumors using NSCT and extreme learning machinery (ELM).

Fusion strategies in classifying COVID-19 CXR images

Different approaches are recently used to classify CXR images with fusion strategies to enhance, detect, and recognize the COVID-19 cases easier and precisely. The fusion of deep learning and statistical features of the enrolled CXR images are performed to ensure the clarity of the relevant information without losing more details where the patches of CXR images of COVID-19 cases are located.

Pereira et al. (2020) presented four phases to classify CXR images: the feature extraction, the Early Fusion technique, the data resampling, and the generation and classification of outcomes for the multi-class and hierarchical scenarios. They used both the Early and Late Fusion strategy based on recognized texture descriptors and a pre-trained CNN model. The fusion strategy is based on a weighted sum, weighted product, and the enrolled features’ voting strength. They achieved an average F1-Score of 0.65 and 0.89 multi-class and hierarchical classification, respectively.

Deep feature fusion algorithm presented by Wang et al. (2021) was utilized to fuze both individual image-level features and relation-aware features to produce Graph Convolutional Networks (GCN) and CNN, respectively. The extracted features are based on self-created CNN that learn image-level individually. The applied algorithm used to classify COVID-19 cases that assist radiologists in detecting COVID-19 cases rapidly. Commonly, deep learning fusion classifiers provide more encouraging results to detect COVID-19 cases than traditional RT-PCR testing. It made the detection and prediction process more reliable with increased accuracy (Panwar et al., 2020). They used a color visualization approach to make the deep learning model more interpretable and explainable.

Two collaborative stream networks presented by Chen et al. (2020) are used to classify multi-label CXR images based on lung segmentation. A self-adaptive weighted fusion scheme is applied to aggregate the contextual information in both global and lung fields with the mean area under the curve AUC = 0.82. Moreover in Li et al. (2020) used multi-resolution convolutional networks to learn the features and employed four different fusion methods that are CNN, Committee, late, and Full fusion strategies for lung classification and the results obtained are 95.01%, 97.17%, 97.92%, and 98.23% respectively.

Parallel-dilated convolutional neural network (PDCNN) based COVID-19 classification system from chest X-ray images is presented by Chowdhury, Rahman & Kabir (2020) generated features are fuzed into the CNN network to produce the final prediction. They used 2,905 chest X-ray images representing COVID-19, Normal, and Pneumonia cases with a reasonable accuracy reached to 96.58%.

COVID-19 hybrid classification approach based on a fusion of CNN and swarm-based feature selection algorithm is presented by Sahlol et al. (2020). This combination is helpful to obtain high performance with minimum computational time. They used fractional order-marine predictors algorithm (FO-MPA) as an optimizer to select the most significant features from deep features produced from CNN that usually have redundancy; therefore, thereby depreciating the resources’ capacity higher classification rate of COVID-19 X-ray images are achieved. The major limitation is eliminating the CNN redundancy, and the low quality of the fuzed image may produce an error in diagnosis and classification. For this reason, in diagnosis and classification issues, the need to improve the quality of the fuzed image is required, and it is conducive to detect the relevant features of the applied images.

One of the well-known methods is Non-subsampled Contourlet Transform (NSCT). Xinqiang, Jiaoyue & Gang (2017) present an image fusion method based on local neighbor features and NSCT with a promising fusion effect on multi-focus images, especially medical images with infrared and visible light images.

Therefore, in this work, we exploit the advantages of deep learning approaches with the NSCT method to obtain more precise and accurate images with specific detail to diagnose CXR COVID-19 cases. The best of our knowledge is the fusion of NSCT, and deep learning features are not used in the CXR COVID-19 classification issue. In Table 1, we investigate the summary of the related work.

Table 1 Summary of the related work.

Author	Modality	Methodology	Discussion	
Wang et al. (2021)	COVID-19 Images	GCN + CNN	The fusion of both individual image-level features and relation-aware features to produce Graph Convolutional Networks (GCN), and CNN respectively	
Wang et al. (2020)	kurtosis map	Hybrid decomposition of NSCT and morphological sequential toggle operator (MSTO)	Their methodology extracted major feature information of the source images and preserved the unambiguous edges with a little produced noise in both visible and infrared image fusion	
Chandra et al. (2020)	CXR images	gray level co-occurrence matrix (GLCM)	extract shape feature from CXR images based on gray level co-occurrence matrix (GLCM) with an improved abnormality detection	
Bhandary et al. (2020)	CXR images	CNN and PCA	The combination of one-dimensional feature vectors and the dimensionality reduction is performed using PCA are then applied to the source CXR images and tested for the normal bacterial pneumonia	
Ozkaya, Ozturk & Barstugan (2020)	CT images	DCNN	pre-trained CNN to fuze different subsets and transfer learning classifiers to classify COVID-19 cases	
Haskins, Kruger & Yan (2020)	MR, CXR, and CT modality.	DCNN	To visualize different registration of essential data of the source images using fixed and moving data labels as well as fixed, and moving images	
Lin et al. (2020)	CT, MR, and PET scan	stacked convolution neural network (DSCNN) for multi band images	DSCNN and NSCT fuze multiband images reconstructed by long short-term memory (LSTM) and DSCNN to overcome the data-driven approach's controllability problem.	
Huang et al. (2020b)	MRI-CT	NSCT and DCNN	(NSCT) by which the fusion technique able to avoid the spectral aliasing and provide more characteristic of the invariance translation	
Maharjan et al. (2020)	Brain Tumor CT	ELM and NSCT	Detect brain tumor using NSCT and extreme learning machinery (ELM].	
Pereira et al. (2020)	CXR images
COVID-19	multi-class
hierarchical CNN	The fusion strategy based on weighted sum, weighted product, and the voting strength of the enrolled features. They achieved average F1-Score of 0.65, and 0.89 multi-class, and hierarchical classification respectively.	
Panwar et al. (2020)	Chest X-ray and CT-Scan images of COVID-19 cases	deep learning and grad-CAM based color visualization	Color visualization approach to make the deep learning model more interpretable and explainable	
Chen et al. (2020)	Multi-label CXR image	self-adaptive
weighted fusion scheme	Contextual information in both global and lung field with the mean area under the curve AUC = 0.82	
Chowdhury, Rahman & Kabir (2020)	2905 chest X-ray images
COVID-19	PDCNN+CNN	Accuracy = 96.58 for COVID-19, Normal, and Pneumonia cases	
Bashir et al. (2019)	CXR, CT, and MRI	multimodal imagery based on using SWT and principal component analysis (PCA)	A dimensionality reduction is performed using PCA and then SWT to extract features	
Lee, Lee & Kang (2018)	CXR images	Poisson-Gaussian noise analysis	NSCT in fusion strategy, the noise distribution produced by CXR images	
Rajalingam, Priya & Bhavani (2018)	CXR and CT	NSCT and DCNN	Hybrid multimodality medical image fusion are applied in both CXR and CT images	
Li et al. (2017)	MRI and PET images	Multi-scale transformation coefficients to produce a fuzed image with inter-scale correlation	They apply MRI and PET images to observe the objectives of fusion performance and determine the source images' miss-registration	

Proposed Algorithm Framework

The system proposed in this article is two-layer image fusion using deep learning as shown in Fig. 1. The proposed method can adaptively decompose two images or more and reconstruct the new image with a high-quality image in the fusion. Using NSCT to decompose the input images to get their high frequency and low-frequency images, and extract their features vector for each low-pass subband and high-pass subband by the CNN-VGG19, combine them our fusion method (NSCT) to achieve the final fusion images.

Figure 1 The proposed algorithm framework.

The system is made up of five major stages, as shown in Fig. 2:

Figure 2 Proposed perceptual two layer image fusion using deep learning.

Data preprocessing: reading CXR images Dataset in grayscale, converting to RGB, resizing, and denoising are all done in the first stage of data preprocessing.

NSCT decomposition: image X and Y or more than two ready images are decomposed into their low-pass subband and high-pass subband images, respectively.

Deep learning convolutional neural networks (VGG19): The third stage is DL using CNN-VGG19 as a feature extractor.

Fusion rule: A temporal consistency of extracted features, Euclidean distance and weights subband calculations are used

NSCT fusion technique: finally, the fuzed image is computed from the fuzed high-pass subband and the fuzed low-pass subband images by applying the inverse NSCT.

Materials and Methods

Dataset preprocessing

The first step before the fusion model is data preprocessing, which includes the following steps:Initially, we start reading the datasets.

All datasets of CXR images in grayscale are converted to RGB images to be appropriate for CNN-VGG19.

One of the significant phases in the data preprocessing is resizing the resulting RGB images. Since the dataset is collected from different waves, they have different sizes aligned into (224, 224).

An aligned CXR image’s appearance is enhanced using the proposed method (Kraetschmer, Dagel & Sanders, 2008).

Finally, the total variation for image components is denoised using the method proposed (Chambolle et al., 2010).

Figure 3 indicates dataset preprocessing steps by taking an example of CXR COVID-19 image cases. The resulting histograms of distributed pixels in each step of the preprocessing reflect that the preprocessing strategy aims to maintain the pixel distribution’s original essence, thus suppressing the abnormal strengths.

Figure 3 Example of dataset preprocessing steps of raw CXR COVID-19 image.

(A) Input CXR COVID-19 image, (B) resized CXR COVID-19 image, (C) refined CXR COVID-19 image, and (D) denoised CXR COVID-19 image.

Fusion based on NSCT

NSCT has significant features of avoiding spectral invariance in aliasing and translation. The decomposition and reconstruction procedure preserves the source image's specifics so that the image’s features can be extracted. NSCT carries out processing on the source image to obtain low-pass frequency and high-pass frequency in each direction, and by inverse NSCT a fuzed image is transformed as shown in Fig. 4 (Huang et al., 2020b).

Figure 4 The NSCT fusion method.

High-pass subbands fusion rule

The CXR is processed as the input image to the NSCT decomposition level stage; therefore, a fusion process integrates the trained image with enhancement performance. High-pass filter fusion’s significant subbands are the augmentation process that performs each source image’s specific features. Equation (1) describes the fuzed high-pass HPF subband image as follows.

(1) HPF(x,y)={HPA(x,y)ifLmapA(x,y)=1HPB(x,y)otherwise

where HPF, HPA and HPB are subband high-pass images for the fuzed image of source IA and IB images, respectively. DmapA(x,y) means the map decision for the high-pass sub-band as determined in Eq. (2).

(2) Dmapi(x,y)={1if⌈Si(x,y)⌉>Q~×R~20otherwise.

In Eq. (2), Si signifies the sliding window with a specific size of Q~×R~, and is concentrated at (x, y) with i number of source images.

Low-pass subbands fusion rule

In Low-pass subbands filter, most source images’ energies are contained to produce significant fuzed images with enhanced performance. While the NSCT filters have the most exhaustive information than the high-pass subbands, there are still restricted decomposition levels of NSCT that cannot filter all the images’ information. Therefore, to ultimately preserve the detailed information of low-pass subbands, we attempt to use different measurements that reflect the fuzed images' structured data based on NSCT (Liu, Liu & Wang, 2015).

Equation (3) investigates the presence of two activity level measures that are implemented to determine the detailed information, which is the weighted sum (WS) of the 8-neighborhood, weighted local energy (WE), respectively.

WE is determined as Eq. (3) as follows:

(3) WE(x,y)=∑m=−rr⁡∑n=−rr⁡Ψ×(m+r+1,n+r+1)×LP(x+m,y+n)2

where LP signifies the low-pass subband of source image at (x, y), WE signify the localized WE at (x, y), Ψ is a matrix that contains (2r + 1) × (2r + 1) and the elements in Ψ are 22r−d. The radius of matrix Ψ is r and d is the distance of four-neighborhood distance to the center of matrix Ψ. A matrix is shown in Eq. (4) investigated that when r is set to 1, the normalized matrix Ψ is

(4) 116[121242121]

Here WE is utilized to measure the structured information, whereas WS is used to measure the detailed extracted features shown in Eq. (5).

(5) WS(x,y)=∑m=−rr⁡∑n=−rr⁡Ψ(m+r+1,n+r+1)×ξ(x+m,y+n)

Ψ is the weighted matrix investigated in Eq. (3), while the ξ is illustrated in Eq. (6). The parameter ξ indicated that full usage of the neighbored information was performed. Therefore, exhaustive information can be restored by ξ. In this scenario, When ξ and WS are achieved, the fusion of both low-pass subband images can be determined by the rule proposed in Eq. (7), given that LPF, LPA, and LPB are low-pass subband images of the fuzed image given the source image IA and IB respectively. WLAN and WSBN are the normalized WS of IA and IB respectively.

(6) ξ(x,y)=|2L(x,y)−L(x−1,y)−L(x+1,y)|+|2L(x, y)−L(x,1,y)−L(x+1,y)|+12|2L(x, y)−L(x−1, y−1)−L(x+1, y+1)|+12|2L(x, y)−L(x−1, y+1)−L(x+1, y−1)|

(7) LPF(x,y)={LPA(x,y)if0.5.WLEAN(x,y)+0.5.WSAN(x,y)≥0.5.WLEBN(x,y)+0.5.WSBN(x,y)LPB(x,y)otherwise

The fuzed coefficients of high-pass subband (HF) and low-pass subband (LF) are determined, the resulting fuzed image (IF) can be acquired by inverting NSCT over {HF,LF}. The inverse transformation of the NSCT is realized by optimizing linear reconstruction for HF and LF based on dual coordinate system operators.

Finally, to obtain a fuzed image, the image is reverse by NSCT. In Fig. 4, a block diagram of the NSCT-based fusion approach is shown.

Convolutional neural network architectures

Recently, the usage in all-purpose of DL algorithms and CNNs has directed several innovations in a diversity of computerized applications, such as object segmentation, classification, and recognition (LeCun, Bengio & Hinton, 2015). DL methods have proven effective in automating learning to represent features and characteristics while actively seeking to remove handcrafted features engineering's repetitive task. By adding a hierarchical layer of feature representation, the DL and CNNs aim to mimic the human visual cortex system’s purpose and construction.

After the 2012 ImageNet competition, CNNs have been commonly used in image processing problems. In a convolution layer, the output feature map is produced when the preceding layer’s feature maps are converted to learnable kernels by using the activation function. Multiple input maps will combine convolutions with each output map. It is generally formulated as it is in Eq. (8).

(8) xjl=f(∑i∈Mj⁡xil−1∗kijl+bjl)

Within Eq. (8), Mj denotes the range of an input map. If both output map j and map k both sum over input map i, then the kernels added to map i are distinct (Liu, Liu & Wang, 2015), for output maps j and k.

Convolutional neural networks for features extraction

Spatial exploitation based on CNN’s

CNN’s consist of a relatively large number of hyperparameters and parameters, such as neurons, number of layers, biases, filter sizes, stride, learning rate, activation function, and weights. Here, different correlation levels can be explored based on other filters as the combinatorial process considers the local area of the input pixels. Different filter sizes encapsulate various complexity levels; small filters typically extract fine-grained while large filters extract coarse-grained data. As a result, researchers exploited spatial filters in early 2000 to enhance quality and explored a spatial filter relationship with network learning. Multiple kinds of research were published in this era indicated that CNN could perform effectively on coarse and fine-grained data when modifying filters.

The technical descriptions of different CNN models, their parameters and principal contribution, rate of error, categorization, and depth are summarized in Table 2 (Khan et al., 2020).

Table 2 Characteristics of CCN used in the proposed framework.

CNN models	Year	Principal contribution	Parameters	Rate of error	Depth	Categorization	Reference	
VGGNet	2014	Homogeneous topology

Using fewer filters

	138 M	ImageNet: 7.3	19	Spatial Exploitation	Simonyan & Zisserman (2014)	
AlexNet	2012	More deep and broader than the LeNet

Using Relu, drop and overlap Pooling

NVIDIA GTX 580 GPU

	60 M	ImageNet: 16.4	8	Spatial Exploitation	Krizhevsky, Sutskever & Hinton (2017)	
ResNet	2016	Residual training

Identity object tracking skip connection

	25.6 M
1.7 M	ImageNet: 3.6
CIFAR-10: 6.43	152
110	Depth and Multipath	He et al. (2016)	
GoogleNet	2015	Introduced principle of block

Divide the idea of transformation and fusion

	4 M	ImageNet: 6.7	22	Spatial Exploitation	Szegedy et al. (2016)	

VGGNet

The experiment in structural technology has accelerated with the active use of CNNs in image classification tasks. A simple and efficient design theory for CNN architectures was proposed by Simonyan & Zisserman (2014) in this regard. Their style, known as VGG, was modular in layer structure patterns compared to AlexNet and ZfNet; VGG was rendered 19 layers deep to visualize the depth with the network’s truly representative capability. ZfNet, the 2013-ILSVRC competition frontline network, indicated that limited filtering could increase the efficiency of CNNs. Based on the results obtained, VGG displaced the 11 × 11 and 5 × 5 filters with a 3 × 3 filter layer stack and demonstrated experimentally that the concomitant positioning of small filters (3 × 3) could produce the impact of large filter sizes (5 × 5 and 7 × 7). By decreasing the number of variables, small filters offer an additional advantage of low computational complexity. These results set a new trend in research for CNN to work with narrower filters. By putting 1 × 1 convolutions between the convolutional layers, VGG controls a network’s configuration, learning a feature vector of the resulting feature maps. For both image classification and localization challenges, VGG provided excellent performance. In the 2014-ILSVRC competition, VGG had been in second place, and due to its simple design, heterogeneous configuration, and improved scale, it gained popularity. The critical drawback associated with VGG is that the use of 138 million variables, making it costly and challenging to implement computationally on low-resource systems (Khan et al., 2020). The graphical representation of VGG19 adopted from Özyurt (2019) is investigated in Fig. 5 and Table 3.

Figure 5 Blocks graphical representation of VGG19.

Table 3 The collected dataset that describe different features related to each patient.

Layer	Patch size/Stride	Depth	Output size	
Convolution	3 × 3 × 64/1	2	224 × 224 × 64	
Max pool	3 × 3/2	1	112 × 112 × 64	
Convolution	3 × 3 × 128/1	2	112 × 112 × 128	
Max pool	3 × 3/2	1	56 × 56 × 128	
Convolution	3 × 3 × 256/1
1 × 1 × 256/1	3
1	56 × 56 × 256	
Max pool	3 × 3/2	1	28 × 28 × 256	
Convolution	3 × 3 × 512/1
1 × 1 × 512/1	3
1	28 × 28 × 512	
Max pool	3 × 3/2	1	14 × 14 × 512	
Convolution	3 × 3 × 512/1
1 × 1 × 512/1	3
1	14 × 14 × 512	
Max pool	3 × 3/2	1	7 × 7 × 512	
Fully connected	–	2	1 × 4096	
Softmax	–	1	1 × 1000	

We used a standard and effective CNN model in this work, named VGGNet, illustrated in Fig. 5, with 16 convolutional and three layers of wholly connected. The convolutional layers' width is comparably small, rising by a factor between 64 in the initial layer to 512, of 2 during each process of max-pooling. There is a reasonable size of 224 × 224 pixels on the input layer. A stride is implemented to retain spatial resolution since each image is transferred through a convolution stack. Pooling is done throughout a fixed window by five max-pooling layers with stride following some but not all convolutional layers. During the first two, three completely connected layers with 4,096 channels are accompanied by a stack of convolution layers with depth varying in various configurations, during the third complete identification.

Temporal consistency is an effective methodology for capturing the contrasting harmony in any input sequence image, especially gray-level images. In turn, for any gray level change in the input sequence, even though at least one must be in the fuzed sequence without any change in contrast or delay. When CXR image sequences are combined, there is a problem reflected in the merging sequence images’ consistency because light stimuli' movement has a responsive effect on the human visual system (Rockinger & Fechner, 1998). The contrast changes introduced by the fusion process will therefore be very distracting, and therefore we will apply (El-Gamal, Elmogy & Atwan, 2016).

A simplified pseudo-code implementation of two-layer image fusion using deep learning is summarized in Algorithm 1.

Algorithm 1 Two layer perceptual image fusion using deep learning.

Procedure: Fusion Schema using NSCT	
Input ← X, Y are two CXR input image(s) and VGGNet is the pretrained VGG-16 network.	
Output ← Z is the fuzed CXR image	
Begin	
 Xa, Xb ← decomposeByNSCT (X)	
 Ya, Yb ← decomposeByNSCT (Y)	
 Za ← DeepLearningFusionRule (Xa, Ya, VGGNet)	
 Zb ← DeepLearningFusionRule (Xb, Yb, VGGNet)	
 Z ← recomposeByNSCT(Za,Zb)	
 saveImageFile(Z)	
 [ ] ← evalautionMetrics(X, Y, Z)	
 Print([ ])	
End	
Procedure Deep Learning Fusion Rule	
Input ← A, B are two subband of CXR input image(s) and VGGNet is the pretrained VGG-16 network.	
Output ← C is the fuzed subband of CXR image	
Begin	
 Afeature ← extractFeatures(VGGNet, A)	
 Avec←Afeature	
 ATC←Avec/norm(Afeature)	
 For each Bi in B subband CXR images	
  Bifeature ← extractFeatures(VGGNet, Bi)	
  Bivec←Bifeature	
  BiTC←Bvec/norm(Bfeature)	
End	
S,Vimg,denom←zeros(B)	
For each Bi in B subband CXR images	
  S←S+BiTC	
  Vimg←Vimg+Bivec	
  denom←denom+Bivec	
End	
 For each Bi in B subband CXR images	
  W←BiTC/denom	
 End	
  C←zeros(B)	
 For each Wi in W	
  C←C+Bi∗Wi	
 End	
End	

Evaluation Matrices

This section is dedicated to exploring the effectiveness of the proposed approach. Two different experimental studies were carried out, discussed, and analyzed in detail due to the variability of the updated standard datasets versions of X-ray COVID-19 images.

All experiments were carried out using the MATLAB 2019b software package running on Microsoft machine with Core i7 processor, 16-RAM, and NVIDIA 4G-GT 740m GPU environment. This section presents a dataset description, validation, and the findings of adding a convolutional deep neural network to NSCT fusion method.

COVID-19 dataset and evaluation metrics

COVID-19 dataset

The Dataset used for this work includes 87 chest X-ray images acquired on 25 cases (17 male, 7 females, and 1 blank) all of which were confirmed with COVID-19. The CXR COVID-19 images cases are available at the Kaggle repository, CXR COVID-19 Dataset (Cohen et al., 2020), existing at https://www.kaggle.com/bachrr/covid-chest-xray.

In this study, a clinical dataset for CXR COVID-19 images was utilized for training and validation. This Dataset consists of images for 25 patients, it has two images or more for each patient. Figure 6 shows four samples CXR images from Dataset for one patient (Bachrr Kaggle, 2020, https://www.kaggle.com/bachrr/covid-chest-xray). Table 4 investigated the complete datasets, including patient data and class labels, and can be demonstrated as follows.

Figure 6 (A–D) CXR COVID-19 images of a 53-year-old patient with pneumonia after 10 days of infection.

Table 4 Evaluation indicator data of various patients (source and fuzed CXR COVID_19 images).

Class labels	Patient data	
Patient ID	Internal Patient Identifier	
Offset	Is very useful to provide as there are several images for the same patient to track progression while being imaged, the number of days after the beginning of symptoms for each image	
Gender	Blank, Male, or Female	
Age	Patient age in Years	
Result	Pneumonia?	
Survival	Have they survived? Yes or No	
Sight	PA, AP, or L for CXRs	
Modality	CT, CXR, or something else	
Date	Date of the acquisition of the image	
Position	Relevance from right to left (hospital name, area, state, country)	
Medical Notes	In specific, about the radiograph, not objective patient	

Evaluation metrics

Performance Analysis needs to be evaluated using a consistently approved standard of image fusion quality. QMI (Quality Mutual Information), Standard Deviation (STD), Peak Signal to Noise Ratio (PSNR), Structural Similarity Index Measure (SSIM), QAB/Fmetric, and Spatial Frequency (SF) are the evaluation measures used and were applied as follows (Xydeas & Petrovic, 2000; Yang et al., 2008; Hossny, Nahavandi & Creighton, 2008; Chen & Blum, 2009; Chen, Pan & Han, 2011).

Standard deviation

To measure the global divergence of the fuzed image, the standard deviation is practically used. Furthermore, the difference between the data obtained and the mean is calculated using it. More useful information from the fuzed image is obtained when the STD value is higher, as investigated in Eq. (9).

(9) σ2=∑i=1I⁡∑j=1J⁡(f(i,j)−μ)2MN

where (I) and (J) are the length and width of the fuzed image f (i, j), and is generally determined as 256 with the mean value of the merged image (μ).

Quality mutual information

Generally, MI is the degree of dependance amongst two source images (X, Y). MI investigated the amount of calculated information that represents the source image concerning the fuzed image. The MI denoted by (M) is relative to the fuzed message by which the formula (M) can be defined as Eqs. (10)–(14):

(10) M=I(x,f)+I(y,f),

(11) I(X,Y)=∑y∈Y⁡∑x∈Y⁡p(x,y)logp(x,y)p(x)p(y)

where p(x) and p(y) represents the Probability Density Functions (PDF) of the two images, and p (x, y) represents the Joint-Probability Density Function (JPDF) of the source image X, Y, and fuzed image.

To estimate the dependability between the random variables X, and Y, the I(X, Y) can be determined as in Eq. (12):

(12) I(X,Y)=∑y∈Y⁡∑x∈Y⁡p(x,y)logp(x,y)p(x)p(y)=0

Given that X, Y and F are the histogram normalization of the source images x, y with the resulting fuzed image f, respectively. By applying MI, the problems regarding the boundness of the metric since is realized as in Eq. (13):

(13) I(X,X)=H(X)

Hence:

(14) QMI=2[I(F,X)H(F)+H(X)+I(F,Y)H(F)+H(Y)]

where H(X), H(Y) and H(F) represents the entropies of X, Y and F, respectively.

Peak signal to noise ratio

It is well-known that PSNR is a quantitative indicator depending on Mean Square Error (MSE). The large value of PSNR leads to improve the fuzed image and enhancement of SNR of the source image Eq. (15).

(15) PSNR=10×log10(L2RMSE2)

Given the PSNR denotes the maximum gray pixel value of the fuzed image, which is 255. The RMSE can be determined as in Eq. (16) by which it represents the difference between the source images and the fuzed images.

(16) RMSE=∑m=1M⁡∑n=1N⁡[ground(m,n)−fused(m,n)]2M×N

RMSE reflects the fuzed image’s ability compared with Ground (m, n) to determine the error with the applied length and width of the image with size M and N, respectively.

Structural similarity index measure

One of the essential benchmarks for the similarity evaluation of the fuzed and source images are SSIM by which the structural similarity (SSIM) metric of the corresponding regions is determined as in Eq. (17) as follows:

(17) SSIM(x,y|w)=(2w¯xw¯y+C1)(2σwxwy+C2)(w¯x2+w¯y2+C1)(σwx2+σwy2+C2)

Which can be decomposed as

(18) SSIM(x,y|w)=(2w¯xw¯y+C1)(2σwxwy+C2)(σwxwy+C3)(w¯x2+w¯y2+C1)(σwx2+σwy2+C2)(σwxσwy+C3)

Given that the parameters C1, C2 and C3 represent the small constants such that C3 = C2/2, and the wx denotes the sliding region in x, so that w¯x is the mean of x, σwx2 and σwxwy represents both variance and covariance of the x and y, respectively.

Spatial Frequency

SF determines the sharpness of the image in fusion. Besides, SF is calculated as the change rate in the gray level of the image. As in Eq, (19), the greater the SF, the higher the image quality.

(19) SF=RF2+CF2

Moreover,

(20) RF=1M(N−1)∑i=1M⁡∑j=2N⁡(X(i,j−1)−X(i,j))2

(21) CF=1(M−1)N∑i=2M⁡∑j=1N⁡(X(i,j)−X(i−1,j))2

where RF and CF represent the row and column frequencies of the image, respectively.

QAB/F measurement

We used the QAB/F parameter determined by the Sobel edge detection operator to evaluate the amount of edge information in the fuzed image compared to the source images. The higher value of QAB/F denotes, the extra data is renewed from the source image, and the edge information is improved and conserved. Generally, the great edge strength produce a great impact on QAB/F as in Eq. (22)

(22) QAB/F=∑n=1N⁡∑m=1M⁡(QA(n,m)WA(n,m)+QB(n,m)WB(n,m))∑n=1N⁡∑m=1M⁡(WA(i,j)+WB(i,j))

where QA(n, m), QB(n, m) is the edge information storage value; WA(n,m),WB(n,m) is the weighting map.

Experimental Results and Discussion

Statistical analysis

In this article, we determine the mentioned parameters statistically in the previous section. These parameters include the average, standard deviation, min, max, and median of the fuzed features obtained in the training phase. Different standard image fusion quality performance metrics including QMI, STD, PSNR, SSIM, QAB/F, and SF are used for evaluation and analysis a statistical study which was applied as follows (Chen, Pan & Han, 2011; Hossny, Nahavandi & Creighton, 2008; Xydeas & Petrovic, 2000; Yang et al., 2008; Chen & Blum, 2009).

Table 5 indicates the performance metric values in terms of QAB/F, QMI, PSNR, SSIM, SF, and STD. Various experiments are performed independently for different traits of the fuzed CXR COVID-19 images. From this Table 5, the following remarks could be concluded as follows:

Table 5 Fused images evaluation metrics of different algorithms.

Statistical measure	Performance metrics	
QAB/F	QMI	PSNR	SSIM	SF	STD	
Average	0.54	0.56	16.92	0.80	0.68	0.21	
STDEV.S	0.05	0.07	2.48	0.03	0.20	0.02	
STDEV.P	0.05	0.06	2.29	0.03	0.18	0.02	
Min	0.49	0.48	13.78	0.76	0.30	0.18	
Max	0.61	0.65	20.91	0.84	0.95	0.23	
Median	0.53	0.58	17.75	0.81	0.68	0.21	

Higher values of PSNR are produced at the maximum level (Max value is 20.91); consequently, the (average value is 16.92).

The Standard Deviation (STD) is determined in both sample STDEV.S and population STDEV.P, and the result of the fuzed CXR COVID-19 images are in between 0.02 to 0.23 maximum value.

These results indicate that the proposed fusion strategy is stabled during the training process as the statistical balancing in the obtained results are achieved.

Performance comparison of the recent CNN architectures of different categories

Six well-known convolutional neural networks were used to study the effect of feature extraction and extraction time: AlexNet, VGG-16, VGG-19, GoogleNet, ResNet-50, and ResNet-101. The best performance was achieved among all CNN networks by VGG-19, as shown in Fig. 7. VGG19 has the smallest value, 512 feature vector length, with a minimum extraction time of 0.543489 sec. Therefore, VGG19 is used in the proposed framework to satisfy the trade-off between feature vector length and extraction time.

Figure 7 Comparative results of different CNN and the proposed framework CNN-VGG19.

(A) Feature vector size vers pre-trained models, and (B) feature extraction time (sec) vers pre-trained models.

Evaluation performance of CXR COVID-19 images for image fusion using DL

CXR imaging characteristics

CXR entering the entity can be sensed in the CXR imaging process, such as Compton scattering effect and other effects to create CXR attenuation, as the thickness or density of the object to be measured varies, the energy attenuation is also diverse. Thus, screening or film creates a CXR image with black and white contrast, with a gradual increase of the CXRs’ dose in penetrating objects, and an increase in the image region's brightness. Once the tube voltage is constant, the tube current increases, the number of photons in the scan range increases, and the entire CXR image’s grayscale increases (Chen, Pan & Han, 2011).

Figures 8 and 9 show images and histogram distribution for CXR COVID-19 images input and fuzed CXR COVID-19 images with a diverse radiation source dose. In the image’s background region, when the gray scale difference is small, then the histogram of this image appears at a single peak and the gray scale near one as shown in Fig. 8C or the gray scale near zero as shown in Fig. 9C. Whereas in the fuzed CXR COVID-19 images, the gray scale value is high in the overall image, and the histogram background region is spread over the grayscale as shown in Figs. 8D and 9D. In our patients’ results, the CXR COVID-19 images had high contrast and many features.

Figure 8 Example of a female 52-year-old patient (P1) CXR COVID-19 image fusion.

(A) Source of CXR COVID-19 images of a female year’s patient, (B) fused CXR output COVID-19 image of a female 52-year-old patient, (C) histogram of CXR source COVID-19 images of a female 52-year-old patient, and (D) histogram of fused CXR COVID19 image of a female 52-year-old patient.

Figure 9 Example of a male 67-year-old patient (P1) CXR COVID-19 image fusion.

(A) Source of CXR COVID-19 images of a male 67-year-old patient, (B) fused CXR output COVID-19 image of a male 67-year-old patient, (C) histogram of CXR COVID-19 source images of a male 67-year-old patient, and (D) histogram of fused CXR COVID-19 image of a male 67 years patient.

Table 6 includes the fuzed CXR COVID-19 image. The value is denoted in bold lettering, which presents the evaluation parameters of the different datasets of the various random patients (P1, P2,…, and P7) of input and fuzed CXR COVID-19 images. Two metrics are used for measurable performance evaluations, such as SF and STD, to assess the proposed algorithm performance. The larger values of the SF for the fuzed image in all patients indicates the higher image resolution. For fuzed CXR COVID-19 images, the highest STD value is calculated. Then without distortion, the proposed algorithm can maintain the CXR COVID-19 image, the fuzed image is more apparent than input images, and the effect of fusion is powerful.

Table 6 Evaluation indicator data of various random patients.

Where the best results are highlighted in bold.

Evaluation parameters		Patients	
		P1	P2	P3	P4	P5	P6	P7	
SF	Input image X	0.6157	0.608	0.6022	0.6557	0.2608	0.8374	0.6123	
Input image Y	0.4209	0.7012	0.6519	0.5346	0.2251	0.6198	0.64725	
Fused CXR COVID-19 image	0.668	0.7944	0.7016	0.6608	0.2958	0.9548	0.6822	
STD	Input image X	0.1563	0.162	0.1921	0.1948	0.2378	0.252	0.15	
Input image Y	0.204	0.1496	0.2184	0.1599	0.2109	0.19	0.1692	
Fused CXR COVID-19 image	0.274	0.216	0.2544	0.2073	0.2451	0.2548	0.2489	

In Fig. 10, the proposed CXR COVID-19 image fuzed algorithm produces CXR COVID-19 images for random seven patients during seven experiments using only two CXR COVID-19 images. Our proposal generates a CXR COVID-19 image with SF greater than or equal to SF of the input image(s). When there are more than two CXR COVID-19 images, we compromise the highest and lowest resolution image to guarantee our proposal’s best performance.

Figure 10 Comparison between input image X and fuzed image of evaluation indicator SF.

Figure 11 shows that the generated CXR COVID-19 image gains a significant SF value against the first input CXR COVID-19 image values. Since the SF measures the sharpness of objects in the CXR COVID-19 image and gray level change ratio, the proposed fusion method leads to better judgment and diagnosis of the patient from his CXR COVID-19 image(s).

Figure 11 Comparison between input image Y and fuzed image of evaluation indicator SF.

A comparative evaluation is performed for the two input CXR COVID-19 image and the produced CXR COVID-19 image based on the STD evaluation metric during seven experiments for random seven patients. The proposed algorithm generates significant STD values five times higher than the two input images vs the same STD values as shown in Fig. 12. We can compromise that the proposed fusion method is efficient enough to generate CXR COVID-19 images that are more useful for the examiner for exploring patient status.

Figure 12 Comparison between input image X, Y and fuzed image of evaluation indicator STD.

Comparative analysis

Validation performance using evaluation metrics

The algorithms for evaluation performance, by comparison, are GFF (Li, Kang & Hu, 2013), MSA (Du et al., 2016b), NSCT + LE (Zhu et al., 2019), NSST + PAPCNN (Yin et al., 2018), respectively. Six metrics are practical to objective evaluation metrics measurements, such as QAB/F, QMI, PSNR, SSIM, SF, and STD (Du et al., 2016a), to determine the competence of the various multi-modal MIF algorithms mentioned above.

In Table 7, The highest QAB/F value is the GFF algorithm. The result is that the GFF algorithm protects the input image's edge information. The GFF algorithm also achieves good results on the evaluation metric for the SF. This algorithm enhances the resolution of the fuzed image by utilizing guided filtering to display each pixel’s saliency and spatial accuracy. The algorithm NSST + PAPCNN performs best on the indicator STD of evaluation, leading to purer fuzed images.

Table 7 Fused images evaluation metrics of different algorithms.

Where the best results are highlighted in bold.

Fusion method		Evaluation metrics	
		QAB/F	QMI	PSNR	SSIM	SF	STD	
Spatial domain	GFF (Li, Kang & Hu, 2013)	0.5978	3.127	42.9419	0.4865	31.7488	63.882	
MSA (Du et al., 2016a)	0.3038	2.8495	28.75	0.4829	17.0158	55.9475	
Transform domain	NSCT + LE (Zhu et al., 2019)	0.5184	2.6052	26.0083	0.4861	31.337	75.5464	
NSST + PAPCNN (Yin et al., 2018)	0.5206	2.6297	25.2629	0.4914	31.7002	77.6748	
Deep Learning	Proposed Algorithm NSCT + CNN − VGG19	0.5415	3.8606	44.8445	0.6509	31.66083	77.2692	

The proposed algorithm NSCT + CNN_VGG19 also accomplishes good results on the QMI, SSIM, and PSNR evaluation metric measures. The use of DL to extract the features improves the proposed algorithm’s performance, so the NSCT + CNN − VGG19 algorithm achieves the highest in the QMI, PSNR, SSIM. The higher PSNR value means the pixel gray value is higher than the comparative algorithms in the fuzed image. Also, the great benefit of SSIM means that the fuzed image and the input image are structurally identical to other algorithms. The higher values of QMI, PSNR, SSIM, as shown in Table 7 and Fig. 13, mean that the fusion effects of images are strong compared to competitive approaches.

Figure 13 Representation of evaluation index data of different methods of fuzed images.

(A) Comprehensive comparative evaluation regards to (STD, SF, and PSNR), and (B) comprehensive comparative evaluation regards to (QAB/F, SSIM, and QMI).

This study compared the evaluation metrics of the different proposals. Upon prior knowledge of the authors, QAB/F, QMI, PSNR, SSIM, SF, and STD were deemed significant metrics in evaluating the fusion method. In turn, the achieved points of the proposal vs other research was compared as shown in Fig. 14 and found that: Two proposals (MSA (Du et al., 2016b) and NSCT + LE (Zhu et al., 2019)) did not meet any of the evaluation measures.

Yin et al. (2018) scored a total of 33% of the overall evaluation metrics (QAB/F and SF) to evaluate the performance of the algorithm (NSST + PAPCNN).

The proposed research achieved 50% of the overall evaluation metrics in QMI, PSNR, SSIM

Figure 14 Cluster dendrogram hierarchy evaluation of the proposed algorithm with competitive metrics of fuzed methods for images.

Validation Performance using classification

To assess the performance of the proposed framework, we used accuracy, sensitivity, and specificity. The descriptions of these measures are as follows:

(23) Accuracy=TP+TNTP+TN+FP+FN

(24) Sensitivity=TPTP+TN

(25) Specificity=TNTN+TP

where:

“TP”: true positives cases in an instance of COVID-19 case, “TN”: true negatives cases, significantly negative COVID-19 case, “FP”: false positives cases, and “FN”: false negatives cases are the incorrect categorized by the classifier for COVID-19.

The evaluation of fuzed CXR COVID-19 images for the pre-trained proposed framework image fusion using DL classification was performed for the COVID-19 dataset using deep learning and compared accuracy in recognizing the different systems as shown in Table 8. The dataset that is already used in classification comparison is the same dataset used in Abbas, Abdelsamea & Gaber (2020), Apostolopoulos & Mpesiana (2020) and Luz et al. (2020).

Table 8 Comparison with other CNN architectures.

Where the best results are highlighted in bold.

VGG19 Pre-trained model	Tuning mode	Performance metrics (%)	
Accuracy	Sensitivity	Specificity	
Abbas, Abdelsamea & Gaber (2020)	Shallow (KNN)	93.42	89.71	95.7	
Deep learning (CNN)	94.59	91.64	93.08	
Apostolopoulos & Mpesiana (2020)	Deep learning (CNN)	98.75	92.85	98.75	
Luz et al. (2020)	Deep learning (CNN)	75.3	77.4	50.0	
Proposed framework	Shallow (KNN)	96.93	57.14	99.2	
Deep learning (CNN)	99.04	85.71	99.6	

Decompose, transfer, and compose (DeTraC) model for detecting COVID-19 from CXR images was presented (Abbas, Abdelsamea & Gaber, 2020). They used different pre-trained transfer learning model based on both shallow and deep learning approaches. They tested their method with and without decomposition, and the VGG-19 accuracy is determined in both shallow and deep learning tuning mode, and the resulting accuracy was 93.42% and 94.59%, respectively.

Apostolopoulos & Mpesiana (2020) presented a transfer learning approach to detect and classify COVID-19 cases from normal cases automatically. They used VGG-19 compared with other transfer learning methods, and the achieved results based on accuracy, sensitivity, and specificity were 98.75%, 92.85%, and 98.75%, respectively.

Luz et al. (2020) presented an architecture based on CNN to detect the abnormality caused by COVID-19 using 13,569, 231 trained, and tested cases, respectively. They tested the three-class labels normal, pneumonia, and COVID-19 using the well-known transfer learning approaches VGG-19, VGG-16, and Resnet-15, and the resulting accuracy was 75.3%, 77.0%, and 83.5%, respectively.

As present in Table 8, The pre-trained proposed model using Shallow (KNN) compared with other pre-trained models and achieved results based on accuracy, sensitivity, and specificity were 96.93%, 57.14%, and 99.2%, respectively. The pre-trained proposed model using Deep Learning (CNN-VGG19) compared with other pre-trained models and achieved results based on accuracy, sensitivity, and specificity was 99.04%, 85.71%, and 99.6%, respectively. In turn, classification rates (accuracy and specificity) indicating the result of fusion as enhancement and riching image with extra details as a side effect of classification in CXR vision application.

Due to the insights in terms of average precision, average recall, and accuracy that is mined based on Fig. 15, the proposed fusion method helps either shallow classifier, KNN, and deep learning classifier (CNN-VGG19) in the classification task. In turn, the fusion is a positive initiative for enhancing the classification and aided computer vision models using CXR COVID-19 images.

Figure 15 Confusion matrix of the proposed framework.

(A) Confusion matrix of the pre-trained proposed framework using VGG19 for CXR COVID-19 images and (B) confusion matrix of the pre-trained proposed framework using KNN for CXR COVID-19 images.

Conclusion

MIF growth varies from the spatial domain, transformation domain, to DL. Its rapid growth also suggests a strong demand for computer-aided clinical diagnosis. Different researchers suggest various fusion methods, each of which has its advantages in the multiple measures of evaluation indicators. However, there exist approximately thirty types of evaluation indicators for MIF. Furthermore, the fusion techniques developed depend on early methods. The researchers enhanced the existing problems in fusion but did not resolve it; distortion of colors and features extraction. In MIF, innovative algorithms remain a major challenge in this field of research.

DL has strengthened the effect of fusion, but research also has some rules; DL structure, for instance, is single, and the amount of training data is limited. Because professional labeling by medical experts is required for the qualified CXR images, they hardly work, and the cost is high. Therefore, there is also a lack of data for training, eventually leading to overfitting.

A fast and effective diagnostic test or protocol will help achieve appropriate early medical care for COVID-19 patients, helping save many lives worldwide. Finding a rapid and effective diagnostic protocol or test becomes one of the critical priorities. This paper is one step ahead towards implementing Deep Learning-based fusion methods to obtain more informative CXR images for the COVID-19 dataset. It could aid in screening or accelerate the speed of COVID-19 diagnosis. We observe that in the COVID-19 CXR image’s background region, when the grayscale difference is small, then the histogram of this image appears at a single peak and the grayscale near one as shown in Fig. 8C or the grayscale near zero as shown in Fig. 9C. Whereas in the fuzed CXR COVID-19 images, the grayscale value is high in the overall image, and the histogram background region is spread over the grayscale as shown in Figs. 8D and 9D. All the fuzed images show the same features, and it works in both hard and light cases. The fuzed algorithm proposed works well in all cases, and its appearance in the Evaluation indicator data of various random patients, as shown in Tables 6 as an example.

The future trend is the study of DL in MIF, according to the previous section. This research proposes a novel MIF algorithm based on DL for Imbalanced COVID-19 Dataset (NSCT + CNN_VGG19). Thus, hybrid decomposition and fusion of NSCT and CNN_VGG19 as features extractor is also used. The proposed algorithm can determine that the proposed fusion method is efficient enough to generate CXR COVID-19 images that are more useful for the examiner for exploring patient status. The comparative evaluation was performed using two methods; the first method is to determine the pre-trained framework efficiency using evaluation metrics. While the second method is based on classifying the fuzed CXR COVID-19 images using the deep learning approach CNN-VGG19 compared with the state-of-the-art.

A comparison was also undergone to evaluate the different MFI algorithms performance models using six metrics as evaluation measures; these are QAB/F, QMI, PSNR, SSIM, SF, and STD. The proposed algorithm NSCT + CNN_VGG19 performed best on the predictor of QMI, PSNR, SSIM evaluation measures. The image was more apparent, and the fuzed image contained extensive information. Finally, the suggested algorithm is more efficient than comparable approaches.

The results demonstrate that the pre-trained proposed framework using fusion based on NSCT with deep learning VGG19 may significantly affect the classification and feature extraction from X-ray COVID-19 images automatically related to the diagnosis COVID-19.

Supplemental Information

Supplemental Information 1 COVID-19 CXR Imageset.

Click here for additional data file.

Supplemental Information 2 Matlab source code for image fusion and evlaution metrics.

Click here for additional data file.

Additional Information and Declarations

Competing Interests

Author Contributions

Data Availability

The authors declare that they have no competing interests.

Omar M Elzeki conceived and designed the experiments, performed the experiments, analyzed the data, performed the computation work, prepared figures and/or tables, and approved the final draft.

Mohamed Abd Elfattah conceived and designed the experiments, performed the experiments, analyzed the data, performed the computation work, prepared figures and/or tables, authored or reviewed drafts of the paper, and approved the final draft.

Hanaa Salem conceived and designed the experiments, performed the experiments, analyzed the data, performed the computation work, prepared figures and/or tables, authored or reviewed drafts of the paper, and approved the final draft.

Aboul Ella Hassanien conceived and designed the experiments, analyzed the data, authored or reviewed drafts of the paper, and approved the final draft.

Mahmoud Shams conceived and designed the experiments, performed the experiments, analyzed the data, performed the computation work, prepared figures and/or tables, authored or reviewed drafts of the paper, and approved the final draft.

The following information was supplied regarding data availability:

The dataset, organized per patient CXR images, is available at Kaggle: https://www.kaggle.com/bachrr/covid-chest-xray.

Code is available in the Supplemental Files.

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
