# Peer review of "A novel perceptual two layer image fusion using deep learning for imbalanced COVID-19 dataset"

_PeerJ Computer Science, doi:10.7717/peerj-cs.364_

## Round 0.1 · original submission · Major Revisions

The presentation of the results must be supported by the statistical analysis section. The network architecture and results must be compared with related work.

Reviewer 1 ·

Basic reporting

The article proposes propose a new method for augmenting a dataset of chest X-ray (CXR) images using a deep learning network combined with perceptual two-layer image fusion. The proposed method is validated on the dataset of images obtained from COVID-19 cases.
Discussion of related work should more focus on the methods specifically applied for COVID-19 CXR image analysis, fusion and classification. Many recent successful methods are not mentioned. Also discuss hybrid methods in which deep learning methods are combined with nature inspired optimization algorithms such as presented by Sahlol et al. (Scientific Reports, 10(1)). How is your approach different from other similar works using NSCT such as described by Xinqiang et al. (2017 2nd International Conference on Image, Vision and Computing (ICIVC), pp. 396–400). You should discuss these and other works in order to provide motivation for your own method. Discuss the limitations of these works and suggest how your approach overcomes these limitations. What is the knowledge gap being filled by this article?
The Methods section should have some formal definition and discussion of the problem solved by this article. What do you mean by image decomposition? Define formally to avoid confusion and misinterpretation.
The dataset preprocessing must be described in detail instead of stating that it “includes many steps”. Describe every step, including any methods used and their parameter values in a way that would allow the replicability of your study.

Experimental design

Why you have selected the VGG16 architecture? Discuss alternatives and support you design choice by proper arguments.
For experimental evaluation of fused images you used image quality metrics, but ultimate evaluation of their usability would be the increased accuracy of disease (COVID-19, in your case) diagnostics.

Validity of the findings

Although your results are compelling, the article should be improved in the following ways:
The presentation of the results must be supported by the statistical analysis section.
Your dataset is very small (just 87 images from 25 cases). The method should be validated on a larger dataset of images, if possible.
Some findings such as “images that are more useful” (L. 560) and “The image was more apparent” (L. 564) are not supported? Who evaluated this “usefulness” and “apparentness” and how?

Additional comments

Visualization of results should be improved.
Figure 1: correct spelling errors (“Hisogram”), some text is garbled (unreadable).
Figure 3: explain all abbreviations in the caption of the figure.
Figure 4 lacks of necessary details: add the size of inputs and outputs of each network layer.
Figures 8, 9: what is the meaning of labels P1 … P7? Explain all labels/abbreviations in the figure caption. What is represented on the y-axis? I assume the value cannot exceed 1, so adjust the range accordingly.
Figure 10: use bars instead of line plot for showing the STD values of fused image.
Figure 11: grouping of metric values by method/algorithm does not allow clear comparison between metric values. I suggest to group by metrics, while color is used to discern between methods.
Figure 12 is not clear. That are you trying to convey? Maybe a different type of diagram would be more useful?

Reviewer 2 ·

Basic reporting

Language style is acceptable, however I suggest to consult structure of sentences and vocabulary. Formatting of algorithms and tables is not unified, make them all equal and unified for better look. The tile of section on page 9 goes together with algorithm, revise it.

Experimental design

The choice of applied CCN-VGG16 is not justified. Why this structure was selected for experimental research? Hay you tried other models? What kind of filters have been used in your neural network model? How do they work? Can you show some results? How is the histogram used in your research? What is the role of histogram application to evaluate the image?

Validity of the findings

Comparisons to other models are necessary to show result over other approaches, compare them and draw conclusions. Why such nn was selected if also other existing models are available? Please show comparative results over images. Authors may relate the research to: Neural network powered COVID-19 spread forecasting model, A neuro-heuristic approach for recognition of lung diseases from X-ray images, Bio-inspired methods modeled for respiratory disease detection from medical images.

Additional comments

What are conclusions from applied methods? Do you have any special observations regarding covid features? Are the features repeated in al images? How is your model related to hard and lighter cases? Is the efficiency the same for both cases?

Reviewer 3 ·

Basic reporting

Paper is well organized and but need considerable improvement for acceptance in “PeerJ”, in current format.

Experimental design

NA

Validity of the findings

NA

Additional comments

This paper proposes a system about novel perceptual two-layer image fusion using to predict COVID-19. The authors discussed, they used CNN for features extraction and fusion method. The author claims that the proposed method have good results on COVID-19 patients X-ray images.
There are some observations, corrections and suggestions regarding this paper.
• In NSCT decomposition, describe properly what is image X and Y.
• Proposed model is not presented properly, should be improved and polished to make it more easily understandable.
• What is temporal consistency in fusion? Describe properly.
• What is the training and testing ratio?
• What is the learning rate of CNN in this work?
• What is the size of VVG feature size?
• What is the size of fusion features size? Length of feature vector?
• How to calculate weight?
• Results are limited. Performance measures should be more. The proposed work should be evaluated with a number of performance measures to prove its validity.
• Figure 1 is not in good quality redraw it.
• In Figure 1
Paper is well organized and but need considerable improvement for acceptance in “PeerJ”, in current format.

---

## Round 0.2 · Minor Revisions

Present motivation for particular architectural decisions when developing your model. Why VGGNet was used?
Improve the presentation of the results:
- Figures 7, 10, 11: consider using a different type of plot: bar plot
- Figure 13: column groups 1 and 4 are not visible.
Present confusion matrix of the classification results and discuss it.

Reviewer 1 ·

Basic reporting

no comment

Experimental design

no comment

Validity of the findings

no comment

Additional comments

In my opinion, the present version of the paper can be accepted for publication.

Reviewer 2 ·

Basic reporting

as stated before

Experimental design

as stated before

Validity of the findings

as stated before

Additional comments

The Authors worked on the paper however my concerns are left the same without any proper answer. There are no justifications for chosen models. We can not see comparative results so it is impossible to evaluate the model in relation to other methods. Authors did not selected nn architecture and gave no descriptive examples. Also features of symptoms are neither presented nor discussed.

---

## Round 0.3 · accepted · Accept

The manuscript is accepted.